# Optimisation of Vitamin B12 Extraction from Green Edible Seaweed (*Ulva lactuca*) by Applying the Central Composite Design

**DOI:** 10.3390/molecules27144459

**Published:** 2022-07-12

**Authors:** Deny Susanti, Fatin Shazwani Ruslan, Muhammad Idham Shukor, Normawaty Mohammad Nor, Nurul Iman Aminudin, Muhamad Taher, Junaidi Khotib

**Affiliations:** 1Department of Chemistry, Kulliyyah of Science, International Islamic University Malaysia, Kuantan 25200, Malaysia; ftnshazwanii@gmail.com (F.S.R.); idhamshukor97@gmail.com (M.I.S.); nuruliman@iium.edu.my (N.I.A.); 2Sustainable Chemistry Research Group (SusChemRG), Kulliyyah of Science, International Islamic University Malaysia, Kuantan 25200, Malaysia; 3Department of Marine Science, Kulliyyah of Science, International Islamic University Malaysia, Kuantan 25200, Malaysia; normahwaty@iium.edu.my; 4Department of Pharmaceutical Technology, Kulliyyah of Pharmacy, International Islamic University Malaysia, Kuantan 25200, Malaysia; 5Pharmaceutics and Drug Translational Research Group, Kulliyyah of Pharmacy, International Islamic University Malaysia, Kuantan 25200, Malaysia; 6Department of Pharmacy Practice, Faculty of Pharmacy, Airlangga University, Surabaya 60115, Indonesia

**Keywords:** seaweed, extraction, vitamin B12, central composite design

## Abstract

Vitamin B12, only found naturally in animal-based foods, is essential for brain functions and various chemical reactions in the human body. Insufficient vitamin B12 leads to vitamin B12 deficiency, common among strict vegetarians due to their limited intake of animal-based foods. Nevertheless, extensive studies have demonstrated that macroalgae, specifically the *Ulva lactuca* species, are rich in vitamin B12 and could be further exploited in future dietary applications. In the current study, the ideal extraction method of vitamin B12 from dried *U. lactuca* was developed and optimised to achieve the maximum vitamin B12 yield. The effects of several extraction parameters, including the solvent-to-solvent, methanol:water (MeOH:H_2_O), and solute-to-solvent ratios, and pH on the total vitamin B12 content were analysed through a two-level factorial and central composite design. The highest vitamin B12 content, particularly cyanocobalamin (CN-Cbl), was recovered through the ultrasonic-assisted extraction (UAE) of oven-dried *U. lactuca* at 3 g:60 mL of solute-to-solvent and 25:75% of MeOH to H_2_O ratios at pH 4. The extraction of CN-Cbl from oven-dried *U. lactuca* that employed the UAE method has elevated CN-Cbl content recovery compared to other extraction methods.

## 1. Introduction

A healthy diet reduces the risk of several diseases, such as heart disease and malnutrition, improving the well-being of individuals. However, nutritional deficiency is often observed among the elderly who might have suffered from nutritional anaemia, including magnesium, folate, and vitamin B12 (cobalamin) deficiencies [1]. Recently, vitamin deficiencies, including that of vitamin B12, were reported to have affected individuals of all ages, including pregnant and lactating mothers worldwide [2]. Strict vegetarians are even more vulnerable to vitamin B12 deficiency, as they limit their consumption of meat products [2]. The frequency of vitamin B12 deficiency among vegetarians was approximately 62%, while 25 to 86%, 21 to 41%, and 11 to 90% were evaluated in pregnant women, children, adolescents, and elderly, respectively [3]. Therefore, the issue needs to be taken seriously to reduce the number of people suffering from vitamin B12 deficiency.

Vitamin B12 (Figure 1) represents all potentially active cobalamins. Cobalamins are a group of cobalt-containing compounds (corrinoids) with a lower axial ligand containing cobalt-coordinated nucleotide (5,6-dimethylbenzimidazole as a base). There are various forms of vitamin B12, namely cyanocobalamin (CN-Cbl), hydroxocobalamin (OH-Cbl), adenosylcobalamin (Adl-Cbl), and methylcobalamin (Me-Cbl). The various forms of vitamin B12 are differentiated based on the β-axial ligand occupancy of the corrin ring [4]. Anaerobic bacteria naturally synthesise vitamin B12. Animal-based foods also contain vitamin B12.

Vegetarians and individuals with vitamin B12 deficiency are required to supplement their diet. For instance, vitamin B12 could be acquired through supplements or food fortified with vitamins, such as nutraceuticals extracted from natural resources such as seaweed, specifically *Ulva lactuca*. Seaweed-based foods exhibited potential in reducing the risks of several diseases, including nutritional deficiency [6], since they are low in calories and high in vitamins, minerals, bioactive metabolites, dietary fibres, and polysaccharides [7].

Wahlstrom et al. [8] stated that the *Ulva* species, particularly *U. lactuca*, is widely known for their edible properties due to their high nutritional value, since they contain high levels of polysaccharides, proteins, vitamins, and trace mineral contents. Furthermore, *U. lactuca* displayed antioxidant, antimicrobial, antiviral, and anti-inflammatory properties [9]. Therefore, the seaweed may exhibit potential as an alternative source of vitamins, especially for the elderly and strict vegetarians. In another report, the green edible seaweed was suggested as a source of vitamin B. A daily intake of 1.4 g/day of *U. lactuca* was sufficient to meet the daily requirements of vitamin B12 [10]. Therefore, in the current investigation, the potential of *U. lactuca* as a source of vitamin B12 was evaluated.

Currently, studies are conducted to identify the best drying processes, extraction methods, and optimised parameters in extracting a high quality and quantity of vitamin B12 from *U. lactuca*. The present investigation compared the extraction methods of vitamin B12 from several types of dried *U. lactuca* to obtain a standardised procedure, and the best extraction parameters to extract an excellent quality and quantity of vitamin B12. Three extraction methods, boiling, orbital shaking, and ultrasonic-assisted extraction (UAE), were employed to extract vitamin B12 from the oven-, sun-, air-, and freeze-dried *U. lactuca* under various extraction parameters, including (i) solvent-to-solvent and solute-to-solvent ratios and (ii) pH.

The parameters were analysed and scrutinised individually using a two-level factorial design during the screening procedure. The two-level factorial experiments functioned as excellent screening tools in estimating the overall effects of the main factors and the interactions between the factors [11]. Gottipati and Mishra [12] also highlighted that the factorial design helped examine pre-treatment variations. Subsequently, a central composite design (CCD) based on the response surface methodology (RSM) was employed for optimal parameters. The optimisation of the factors was crucial to develop a standardised extraction method that would produce the desired amount of vitamin B12. The range for each factor examined in the current investigation was varied according to previous reports, which were 25:75 to 75:25%, 3:60 to 3:90 g/mL, and pH 3 to 5, for the solvent-to-solvent and solute-to-solvent ratios and pH, respectively [13,14].

## 2. Results

### 2.1. Screening the Extraction Methods and Drying Conditions

Three extraction procedures, boiling, orbital shaking, and UAE, were employed to extract vitamin B12 from the oven-, sun-, air-, and freeze-dried *U. lactuca* samples. According to the HPLC qualitative analysis, the *U. lactuca* contained CN-Cbl, one of the vitamin B12 analogues. The retention time was recorded at 1.9 min, which corresponded to the retention time of standard CN-Cbl. 

Each extraction procedure of the dried sample was denoted with an abbreviation in the other section.

The highest CN-Cbl content (Appendix A) was observed in the sun-dried *U. lactuca* samples extracted following the orbital shaking procedure (0.0356 mg/mL). The oven-dried samples extracted through the UAE method comprised 0.0236 mg/mL, while the oven-dried samples extracted by the boiling method consisted of 0.0210 mg/mL CN-Cbl. Although the SDO method yielded the highest CN-Cbl content, the model from the two-level factorial was insignificant (*p* > 0.05). Resultantly, eliminating the factors improved the model during the optimisation process. In contrast, the ODU method exhibited a remarkably higher CN-Cbl content (Figure 2) than those extracted employing the boiling (ODB) technique. It has been demonstrated that the mean CN-Cbl content obtained from the UAE method was 43% more than its heat-extracted counterparts [15].

The oven-dried *U. lactuca* samples exhibited a positive and significant effect (*p* < 0.05) on the CN-Cbl yield when extracted under the most suitable parameters and extraction method. Moreover, Dang et al. [16] emphasised that the optimal temperature for oven-drying was 40 °C when drying brown seaweed, *Hormosira banksia*, as rapid degradation of the components occurred when the drying temperature was over 40 °C. The temperature applied in the oven drying method in the present investigation was similar. However, the drying technique should still be cautiously screened and selected beforehand depending on the compound of interest which needs to be extracted. For instance, the oven-drying technique reduced the total flavonoid content yield by 30% and the power of ferric reducing antioxidants by 17% obtained from *Chlorella vulgaris* algae samples [17].

### 2.2. Screening for Significant Factors Using the Two-Level Factorial Design

Apart from determining the suitable extraction and drying methods for *U. lactuca* to obtain vitamin B12, screening for significant factors was also crucial. The ideal experimental design was determined by screening the concerned factors, the solvent-to-solvent ratio (A), pH (B), and solute-to-solvent ratio (C), utilising the two-level factorial design. Appendix A displays the two-level factorial designs with five centre points and the experimental layout representing 21 experimental runs with the concentration of CN-Cbl as the response. The concentration of CN-Cbl obtained from oven-, sun-, air-, and freeze-dried *U. lactuca*, extracted through the boiling, orbital shaking, and UAE methods, are also included. The significance of each model, individual model terms, and lack of fit were determined based on their respective *F*- and *p*-values. The results showed that the ODB, ADB, ODU, and ADU models were significant, with a *p*-value < 0.05, suggesting that the models were fitted adequately. However, referring to the ANOVA analysis presented in Appendix A, the model for ODU (Appendix A) exhibited the highest *F*-value at 76.53, while the model for ODB was 22.26, ADB at 4.57, and ADU was 10.49. The highest CN-Cbl content was also recorded from the ODU model. Therefore, the ODU model was further optimised to interpret the relationship between the significant factors and the response by employing the CCD.

According to the ANOVA analysis (Appendix A), the main effects of factors A and B were significant (*p* < 0.05). The B factor was more significant than A, which exhibited the *F*-values of 49.69 and 10.53, respectively. Moreover, the two-level interactions of solvent-to-solvent ratio and pH (AB), solvent-to-solvent ratio and solute-to-solvent ratio (AC), pH and solute-to-solvent ratio (BC), and the overall interaction (ABC) exerted significant positive effects on the total CN-Cbl yield from the dried *U. lactuca* samples. The findings also suggested that the single interaction of the main effects (A and B) and their interactions (AB, AC, BC, and ABC) influenced the total CN-Cbl extracted.

The effects of an individual factor, C, on the yield of CN-Cbl was insignificant (*p* > 0.05). The observation might be due to the less remarkable statistical relevance in the extraction procedure possessed by the factor [18]. Scientifically, the solvent type and strength are the leading influential elements in the optimisation procedure of an extraction method. Moreover, the *F*-value of the curvature was 60.40 with a significant *p*-value. Hence, the optimisation procedure by CCD was required to approximate the response and explain the relationship between the significant factors and the response from the ODU samples.

### 2.3. Significant Factors That Affected the CN-Cbl Yield from the Oven-Dried U. lactuca Samples That Employed the UAE Technique 

The two-level factorial design of ODU indicated that the solvent-to-solvent ratio and pH demonstrated significant effects over the concentration of CN-Cbl content obtained from the extraction process. Although a single solute-to-solvent ratio exhibited an insignificant effect on the yield, eliminating the effect was not required. The observation was because the solute-to-solvent ratio interacted with other factors in the two, and in their overall interactions. The diagnostic analysis also revealed that the two-level factorial computed an identical distribution of the normal probability of residuals and residuals-versus-predicted-value plots. On the other hand, Figure 3 displayed no outliers in the residual analysis. Consequently, the solute-to-solvent ratio was fixed at the minimum value of 3 g:60 mL in the subsequent CCD procedure. The minimum and maximum values for factor A were within 0:100–50:50%, and 3–5 for factor B. The values were determined based on the desirability ramp, as shown in Figure 4.

### 2.4. Optimising of the ODU CN-Cbl Yield through CCD

The CN-Cbl yield was further optimised using a second-order model, the CCD. It was necessary to approximate the response in this study since the curvature in the initial model was significant (*p* < 0.05). The CCD was applied by augmenting the factorial design with appropriate axial points for the analysis. The α-value of the CCD was set at 1, and 5 centre points were added, generating 13 runs. Table 1 displays the optimised design layout of the CCD for UAE oven-dried *U. lactuca* samples with the concentration of CN-Cbl as the response.

Table 2 presents the ANOVA analysis of the CCD. Only two significant factors, A and B, were optimised during the extraction of CN-Cbl. The solute to solvent ratio was kept constant at 3 g of the dried sample extracted in 60 mL of the solvent. Based on the ANOVA analysis, the generated quadratic model was desirable (*F*-value at 49.26) and demonstrated an insignificant lack of fit. Individual factors A and B remained significant, as well as the two-level interactions between AB, contributing to the concentration of CN-Cbl.

The solvent-to-solvent ratio squared (A^2^) and pH squared (B^2^) were included in the significant model terms after the model was fitted with the second order. According to the normal probability plots of CCD (Figure 5), all the residuals lay close enough to form a straight line, indicating that errors were distributed normally. The residuals in the plot of residuals versus predicted value were structureless, thus demonstrating that the investigated model satisfied the assumptions. Therefore, the fitted quadratic model was correct and desirable. The R^2^ value of the quadratic model was 0.9724, indicating that 97.24 % of the variation in the responses was explained. The predicted R^2^ value, 0.8343, was in reasonable agreement with the adjusted R^2^ value of 0.9526, indicating the adequacy of the quadratic model, as the difference was less than 0.2. The adequacy precision ratio of 16.9051 was also desirable as the ratio was exceeded 4. Consequently, the generated quadratic model was fitted to navigate the design space.

Equations (1) and (2) are the final empirical models in terms of the coded and actual factors, respectively, for the concentration of CN-Cbl (mg/mL).
Concentration of CN-Cbl = 0.0346 − 0.0053 A − 0.0040 B + 0.0037 AB − 0.0091 A^2^ − 0.0139 B^2^,(1)
Concentration of CN-Cbl = (−0.0160010 − 0.000080)A + (0.103123)B + 0.000150(A × B) − 0.000015(A^2^) − 0.013860(B^2^)(2)
where A is the solvent-to-solvent ratio, B is the pH, and C is the solute-to-solvent ratio.

### 2.5. Analysis of the Optimum Region by Employing CCD 

Optimum parameters in extraction procedures play crucial roles in benefiting the yield of an extract. In the CCD, the optimum regions of the parameters were analysed based on the contour plots of the factors. The contour plot of the AB interaction is displayed in Figure 6a, with the solute-to-solvent ratio kept constant at 3 g:60 mL. According to the results, the optimum point, the highest CN-Cbl content, was located at the centre of the contour. The observations indicated that the CN-Cbl content increased when the extraction was performed at 25:75% MeOH:H_2_O and pH 4.

The 3D surface plot of the AB interaction, illustrated in Figure 6b, manifested a curve was with a maximum point at the solvent-to-solvent ratio approximately within 20:80–30:70% and pH 4–4.5. The lowest concentration of CN-Cbl (0.0055 mg/mL) was obtained when dried *U. lactuca* was extracted at 50:50% solvent to solvent ratio and pH 5, followed by 0.0072 mg/mL at 0:100% solvent to solvent ratio and pH 5, and 0.0084 mg/mL at 50:50% solvent to solvent ratio and pH 3. Therefore, 25:75% solvent to solvent ratio and pH 4 were selected as the optimum parameters for the new extraction protocol.

### 2.6. Model Conformation and Validation Test

Experimental runs listed in Appendix A were employed to verify the adequacy of the generated model. The concentration of CN-Cbl content in the oven-dried *U. lactuca* for each run was at a 95% prediction interval based on the model developed in the CCD. All the tested runs exhibited error percentages less than 15% between the actual and predicted mean for the concentration of CN-Cbl content. The overall data mean was also within the 95% prediction interval low and 95% prediction interval high range. Therefore, the developed model was reasonably well fitted and precisely presented the effects of the extraction parameters on the concentration of CN-Cbl content.

### 2.7. Microbiological Assay of Vitamin B12

The purified samples were subjected to microbiological assays that utilised *E. coli* and *L. leichmanii* to confirm the presence of vitamin B12 in the oven-dried *U. lactuca* samples extracted by the UAE method. A zone of *E. coli* and *L. leichmanii* growth were observed surrounding the wells containing CN-Cbl standard and the purified sample. The observations confirmed the presence of vitamin B12, specifically CN-Cbl. There was no growth surrounding the wells of the control, which contained distilled water.

### 2.8. Liquid Chromatography/Electronspray Ionisation-Mass Spectroscopy/Mass Spectroscopy (LC/ESI-MS/MS) Analysis of Vitamin B12

The purified extract containing CN-Cbl compound extracted from oven-dried *U. lactuca* by utilising the UAE method was further analysed and characterised through LC/ESI-MS/MS. Figure 7 demonstrates the presence of M + H CN-Cbl, which was 1355.5232 *m*/*z*.

A similar spectrum was observed in a previous study that analysed the CN-Cbl content extracted from *Spirulina platensis*. The spectrum displayed the presence of M + H CN-Cbl, which was 1356.58 *m*/*z* [14]. The current study results confirmed that extracting vitamin B12 from the oven-dried *U. lactuca* by employing the UAE method enhanced the amount of CN-Cbl recovered.

## 3. Discussion

Vitamin B12 molecules are complex, very reactive, and water-soluble. Therefore, the measurement and extraction of vitamin B12 from food, plant, and algae samples present an enormous analytical challenge. Commonly, the vitamin B12 content in algal materials is low, alongside the presence of other metabolites [19]. The focus of this research was to vary the extraction procedure and drying conditions of vitamin B12 from *U. lactuca* to maximise the yield of vitamin B12. The investigation also introduced novel extraction methods, and reduced the use of hazardous solvents during the extraction of vitamin B12. In addition, in the present study, the extraction solvent was standardised to ease future research. Previous studies discussed that certain organisms might contain the vitamin B12 analogues such as OH-Cbl, Adl-Cbl, and Me-Cbl. However, due to their unstable nature and extraction conditions, the components might be lost during the extraction procedure [14]. Therefore, methanol and water were used in the present study as the extraction solvent since vitamin B12 is a water-soluble vitamin and, owing to the low boiling point of methanol, the decomposition of thermolabile bioactive compounds was prevented [20].

Our data showed that the ODU method exhibited a remarkably higher CN-Cbl content than those extracted employing the boiling (ODB) technique. The same trend was observed in a study on CN-Cbl extraction from a solid dietary supplement that compared the final CN-Cbl contents from UAE and heat extraction methods [15]. Moreover, numerous studies have applied the novel extraction method to extract compounds, such as polyphenols, antioxidants, and prebiotics from macroalgae [16,21]. Our findings agreed that the UAE method that operated at low temperatures enabled the preservation of thermolabile compounds and prevented the structure from being entirely damaged [15,22]. Hence, obtaining *U. lactuca* following the UAE method might be adequate to liberate CN-Cbl prior to quantitative analysis. The procedure could be a standardised protocol in future extractions of CN-Cbl from macroalgae.

The different drying conditions for the extraction procedure of vitamin B12 from *U. lactuca* were examined. The oven-dried *U. lactuca* samples exhibited a positive and significant effect (*p* < 0.05) on the CN-Cbl yield when extracted under the most suitable parameters and extraction method. The findings agreed with the reports from Silva et al. [23], highlighting that oven drying stabilised the *Ulva* species, such as *U. rigida*, during the extraction of specific bioactive components within the optimised temperature. Resultantly, the UAE method was selected as the best procedure in extracting CN-Cbl from the oven-dried *U. lactuca*.

Based on the AB interaction in 3D surface and contour plot, lowering and increasing the solvent-to-solvent ratio and pH beyond the optimum point exhibited no improvement in the concentration of CN-Cbl extracted. The best solvent-to-solvent ratio utilised was 25:75% MeOH:H_2_O. Kumudha and Sarada [14] highlighted that water was the best solvent to extract vitamins since it could extract several water-soluble molecules, including vitamin B12. Other solvents introduced in the extraction of vitamins include water with ethanol and dimethyl sulphoxide. The current study employed water with MeOH as the solvent since MeOH exhibited the highest efficiency as the eluting solvent for vitamin B12 [24]. According to Fang et al. [24], the suggested solvent-to-solvent ratio of 50% MeOH:50% H_2_O was sufficient to meet the requirement. However, 25% MeOH:75% H_2_O was evaluated as the optimum solvent-to-solvent ratio in this investigation, which might be attributed to the different samples used.

The highest CN-Cbl content was recorded at pH 4, indicating pH 4 was the optimum pH to yield the maximum CN-Cbl content from the dried *U. lactuca* samples. A declining trend in CN-Cbl content was observed when the pH was decreased to 3 and increased to 5. According to Chandra-Hioe et al. [15], the extraction of vitamin B12 required specific extraction conditions as it is only stable at lower pH, in the absence of ultraviolet, and the presence of other vitamin B varieties, such as thiamine, niacin, and ascorbic acid. Therefore, the decreasing CN-Cbl yield as the pH was increased to 5 might be because several vitamin B12 compounds were destroyed. Vitamin B12 is unstable under highly acidic and alkaline conditions [25]. Bajaj and Singhal [26] reported that the concentrations of vitamin B12 in an aqueous solution with pH 2 were significantly lower than in solutions with pH up to 6. The observations demonstrated that the degradation rate of vitamin B12 accelerated progressively under a very low and high pH.

Microbiological assay using *E. coli* and *L. leichmanii* led to the confirmation of the presence of vitamin B12 analogues of CN-Cbl compound in the purified sample extract. It was then further analysed and characterised through LC/ESI-MS/MS. Considering these findings, the extraction processes of *U. lactuca* extracts were mainly affected by the extraction method and several important parameters, and these might be a suitable candidate for the novel standardised and simplified extraction process of vitamin B12 from green edible macroalgae. In conclusion, our study demonstrated that *U. lactuca* extracts possessed vitamin B12 analogues, particularly CN-Cbl, which usually only found in animal-based products. The extract also showed some significant effects towards various parameters and extraction conditions. The high content of vitamin B12 analogues, particularly CN-Cbl present in *U. lactuca* extracts, may indicate the suitable conditions and parameters for a new standardised extraction protocol developed in this study.

## 4. Materials and Methods

### 4.1. Sample Collection and Preparation

The *U. lactuca* samples were collected fresh from Merambong Island, Johor, Malaysia, at the coordinates (Latitude 1°18′60.00″ N, Longitude: 103°36′59.99″ E), in January 2020 and January 2021, with water salinity and range temperature of 12 ppt and 25–31 °C, respectively. The species was identified and validated by algae expert, Assoc. Prof Dr, Normawaty Mohammad Nor (Department of Marine Science and Technology, IIUM) during the sample collection. Potential epiphytes were removed from the fresh algae and rinsed on the spot with seawater. Subsequently, the samples were oven-, sun-, air-, and freeze-dried to a constant weight. The dried samples were ground in a grinder for 5 min to obtain a fine and homogeneous powder pre-storage in different sealed bags at room temperature for further investigation.

The drying methods employed are presented as follows:Air drying: The *U. lactuca* samples were placed in a single layer on a clean flipchart paper and air-dried in the laboratory using fan (speed 300 rpm) to keep air circulating until the sample reached a constant weight (temperature 25 °C).Sun-drying: The *U. lactuca* samples were placed in a single layer on a clean aluminium foil and dried under direct sunlight until the sample reached a constant weight (temperature 32 ± 2 °C).Oven drying: The *U. lactuca* samples were placed on an aluminium foil in a single layer and dried in an oven drier (Sanyo, OSA, JP) until a constant weight was obtained. The oven temperature was set to 40 °C.Freeze drying: The *U. lactuca* samples were frozen in a freezer (Sanyo, OSA, JP) overnight at −80 °C and then being placed in a chamber that combines a chilled condenser and a vacuum pump to aid sublimation of water. The freeze-drier (Thermo Scientific, WLM, USA) was set at a cryo-temperature of −50 °C.

### 4.2. Extraction of Vitamin B12

Three grams each of the ground air-, oven-, sun-, and freeze-dried *U. lactuca* samples were added into 100 mL amber Schott bottles. Next, the samples were suspended in methanol (MeOH) (Qrec Sdn. Bhd., KL, MY) and ultra-pure water (Scope-T, CN) solvent mixture at three different ratios, which were 25: 75, 50:50, and 75:25 % of MeOH: H_2_O. The solute-to-solvent ratios were also varied, from 3 g:60 mL, 3 g:75 mL, and 3 g:90 mL, respectively. The experimental design for the different extraction procedures and parameters applied were performed as mentioned in the Appendix A. The extraction of vitamin B12 from the dried *U. lactuca* samples was conducted in the dark since vitamin B12 is light sensitive [27]. The vitamin B12 extracts were centrifuged at 6000 revolutions per minute (rpm) for 10 min (Hettich Zentrifugen, BW, DE). The supernatant was evaporated in a vacuum to procure pure vitamin B12 extracts. The pH of the extracts was adjusted to within 3 to 5, based on the experimental design obtained from the two-level factorial and CCD designs by RSM. The extraction methods utilised are listed as follows:Boiling Extraction Method: The dried *U. lactuca* samples were boiled using hotplate (Thermolyne Thermo Fischer Scientific, Waltham, MA, USA) for 20 min at 65 °C [14]. Upon extraction, the extracts obtained were centrifuged and kept in below 15 °C chiller (Sanyo, OSA, JP) before being purified.Orbital Shaker Extraction Method: The dried *U. lactuca* samples were shaken on an orbital shaker (Thermo Fischer Scientific, WLM, USA) for 30 min [28] at 200 rpm and centrifuged upon extraction. The extracts were kept in below 15 °C chiller before being purified.Ultrasonic Assisted Extraction Method: The dried *U. lactuca* samples were suspended in the solvent mixture and extracted in an ultrasonic water bath (WiseClean, CH) with the power and frequency set to 665 Watt and 20 kHz, respectively, for 30 min. The extracts obtained were cooled and kept in a chiller below 15 °C before being purified.

### 4.3. Purification and Determination of Vitamin B12

The purification of the vitamin B12 extracts was completed by passing the samples containing vitamin B12 through an Amberlite XAD-2 column (Merck KGaA, Darmstadt HE, DE). First, the column was filled with pure MeOH (Merck & Co., Kenilworth, NJ, USA) and an Amberlite XAD-2 resins slurry to 15–16 cm bed height. Next, the MeOH was drained and replaced with deionised water to reach equilibrium. After 15 min, the column was drained, and the vitamin B12 extracts were loaded. Finally, the vitamin B12 was drained very slowly for 3 h and eluted with 80% MeOH to increase the purification efficiency [14]. 

The concentrated solution was spotted on a silica gel thin-layer chromatography (TLC) plate (Merck Millipore, Burlington, MA, USA) and developed with 7:4:5:1:1 of 1-butanol (Merck & Co., Kenilworth, NJ, USA), chloroform (Merck & Co., Kenilworth, NJ, USA), acetic acid (Merck & Co., Kenilworth, NJ, USA), ammonia (Merck & Co., Kenilworth, NJ, USA), and water (H_2_O) solvent in the dark at room temperature. The TLC plate was first rinsed with a dichloromethane (Merck KGaA, Darmstadt HE, DE) and MeOH (Qrec Sdn. Bhd., KL, MY) solution at a 1:1 ratio before being spotted with the vitamin B12 extracts. The spots on the TLC plate were dried, collected, extracted with 80% MeOH, evaporated to dryness under reduced pressure, and dissolved in 50 mL of distilled water [29]. The concentrated solution was further characterised through high-performance liquid chromatography (HPLC) (Perkin Elmer, Waltham, MA, USA).

### 4.4. High-Performance Liquid Chromatography (HPLC) Analysis

#### 4.4.1. Sample Solutions

Vials containing crude extract samples were prepared by diluting the extract with ultra-pure water in 1:2 mL crude extract to ultra-pure water ratio. The vials containing the extracts were vortexed to ensure homogeneity. The dissolved crude extract was filtered using Millipore^®^ SLHN033NB Millex^®^ HN Syringe Filter Nylon Membrane; 0.45 µm in diameter (Merck Millipore, Burlington, MA, USA).

#### 4.4.2. Standard Solutions

Analytical standard stock solutions of vitamin B12 (Merck & Co., Kenilworth, NJ, USA) were purchased from Merck for quantification purposes. The standards consisted of CN-Cbl, OH-Cbl, Adl-Cbl, and Me-Cbl. The stock solutions were diluted with HPLC grade MeOH (Merck & Co., Kenilworth, NJ, USA). The stock solutions were employed to create three to five calibration points for the calibration curve. Each calibration curve for CN-Cbl, OH-Cbl, Adl-Cbl, and Me-Cbl was plotted, individually.

#### 4.4.3. Chromatographic Conditions

The concentrated and purified samples were analysed through a reverse-phase HPLC using Luna^®^ C18(2) column (Phenomenex, Torrance, CA, USA). The samples were eluted with MeOH (A) (HPLC grade (Merck & Co., Kenilworth, NJ, USA) and 0.1% acetic acid (B) HPLC grade (Merck & Co., Kenilworth, NJ, USA) as the mobile phase in an isocratic mode for seven minutes. The injection volume, oven temperature, and flow rate were set at 10 µL, 30 °C, and 0.9 mL/min, respectively. The chromatograms were obtained at a maximum wavelength of 351 nm with an ultraviolet (UV) Flexar PDA Plus detector (Perkin Elmer, Waltham, MA, USA). All data obtained were analysed using Perkin Elmer Chromera software version 4.1.0.6386.

### 4.5. Statistical Analysis

Response surface methodology (RSM) two-level factorial design and Central Composite Design (CCD) were employed to determine the correlation of the factors (MeOH:H_2_O and solute:solvent ratios and pH). The model significance and interactional effects between the factors were observed based on the *F*-test and *p*-values, respectively. The effects were significantly different at *p* < 0.05 and a confidence level of 95%. The two-way analysis of variance (ANOVA) was employed to analyse the model and the effects of the factors. Initially, the two-level factorial design with centre points was applied to screen out the most significant factors. Subsequently, the CCD was used to develop and analyse the optimised extraction procedure model. The computational analysis by the software generated the correlation values.

### 4.6. Microbiological Assays of Vitamin B12 Using Eschericia coli and Lactobacillus leichmanii

*E. coli*, American Type Culture Collection (ATCC) 25922 (Kwik-StikTM, Saint Cloud, MN, USA), a Gram-negative strain, and *L. leichmanii*, (ATCC) 7830 (Kwik-StikTM, Saint Cloud, MN, USA), a Gram-positive strain, were grown in a maintenance medium at 37 °C, mixed with vitamin B12 assay agar, and pour plated. The CN-Cbl standards and purified samples (20 µL) were inoculated into wells bored in the solid agar medium. The plates and distilled water used as control were incubated at 37 °C for 24 h. The zone of growth was observed after 24 h.

### 4.7. Tandem Mass Spectrometry/Mass Spectroscopy (MS/MS) Analysis of Vitamin B12

The electron spray ionisation mass spectrum (ESI-MS) (Bruker, Billerica, MA, USA) was performed in positive mode utilising a linear gradient of MeOH (A) (Merck & Co., Kenilworth, NJ, USA) and 0.1 % acetic acid (B) HPLC grade (Merck & Co., Kenilworth, NJ, USA) as the mobile phase. The Positive-ion tandem mass spectrometry (MS/MS) experiments were performed in product mode on time-of-flight (TOF) mass spectrometer (Agilent 6545). The cone and desolvation gas flow were set at 28 and 1000 L/h.

The samples were introduced into the mass spectrometer through a direct flow injection system through a solvent delivery at a 0.3 mL/min flow rate. The capillary and cone voltages were set at 3.00 kV and 28.00 V, respectively. The source, desolvation, and column temperature were set at 120, 400, and 35 °C, respectively. The mass spectra of the samples were recorded and compared to the mass spectra of other samples reported by previous studies.

## 5. Conclusions

A non-conventional extraction method using ultrasonic extraction (UAE) has improved the extraction yield of vitamin B12, particularly CN-Cbl, from the green edible macroalgae *U. lactuca*. The UAE employed several important parameters, such as solvent-to-solvent, solute-to-solvent ratio, and pH. The highest CN-Cbl was obtained in the sun-dried *U. lactuca* assisted by the orbital shaking procedure (0.0356 mg/mL). The ANOVA analysis showed that low solute-to-solvent ratio, optimised solvent-to-solvent ratio, and enhanced pH increased the recovery of CN-Cbl. The highest CN-Cbl content was recovered when the extraction of oven-dried *U. lactuca* that utilised the UAE method was conducted at 3 g:60 mL solute to solvent ratio and 25:75% MeOH: H_2_O at pH 4. The CN-Cbl content obtained from the oven-dried *U. lactuca* using the UAE technique was elevated compared to the boiling and orbital shaking extraction methods. The optimised parameters would help future research on how to obtain the highest yields of vitamin B12 that would be useful in the nutraceutical industry.

## Figures and Tables

**Figure 1 molecules-27-04459-f001:**
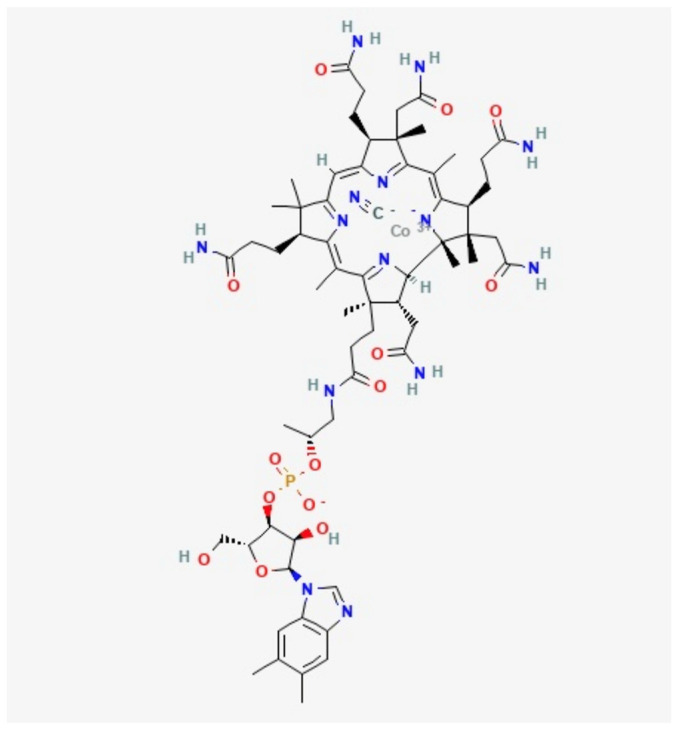
Vitamin B12 chemical structure [5].

**Figure 2 molecules-27-04459-f002:**
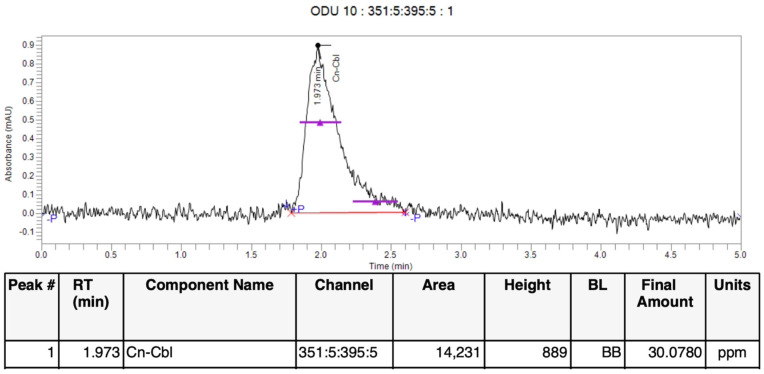
HPLC chromatogram of optimised purified vitamin B12 by CCD, oven dried *U. lactuca* extracted by UAE (ODU) run 10.

**Figure 3 molecules-27-04459-f003:**
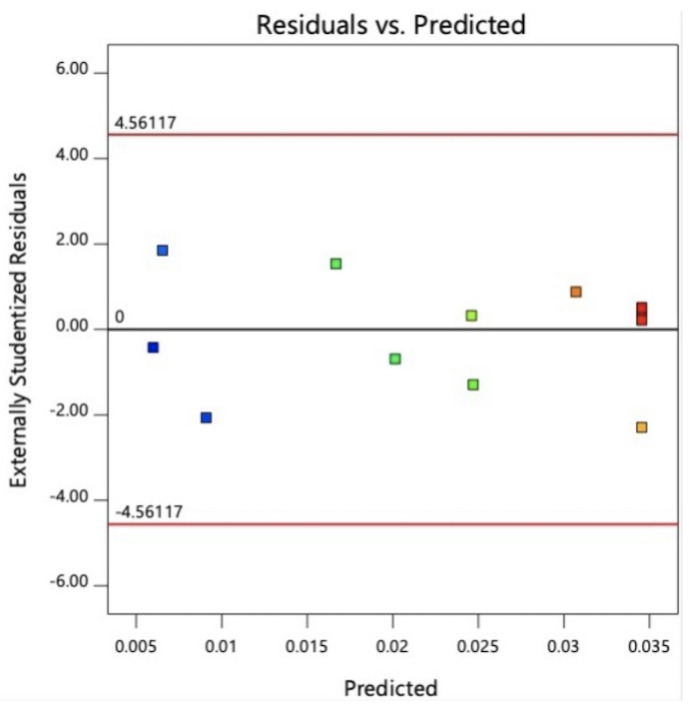
Residual analysis of ODU from 2-Level Factorial.

**Figure 4 molecules-27-04459-f004:**
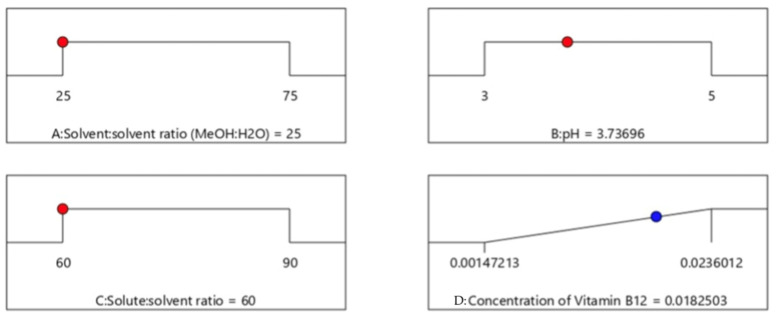
Desirability ramp for numerical optimization of independent variable, solvent:solvent ratio, pH and solute:solvent ration for the concentration of CN-Cbl of ODU sample. A: solvent:solvent ratio (MeOH:H_2_O); B: pH; C: solvent:solvent ratio; D: concentration of CN-Cbl.

**Figure 5 molecules-27-04459-f005:**
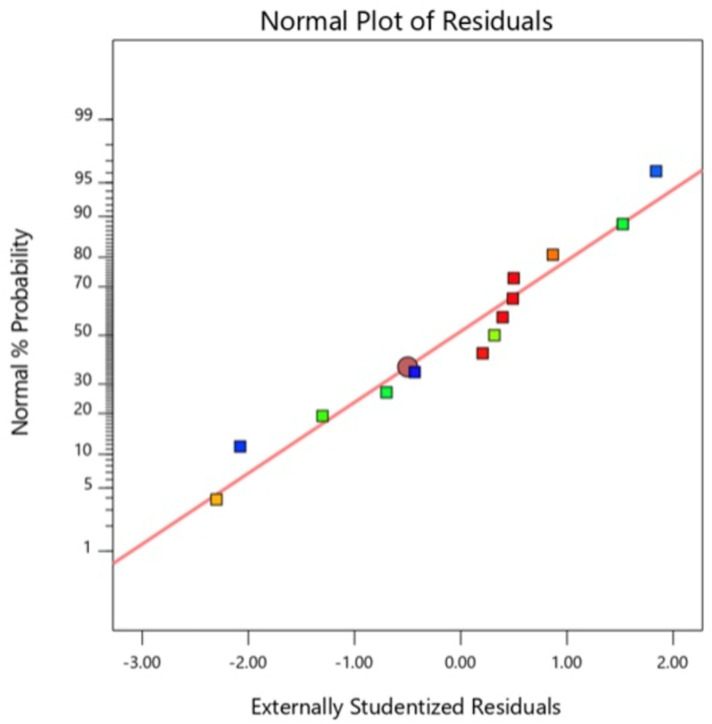
Normal probability plot of ODU sample extracted by UAE method using Central Composite Design.

**Figure 6 molecules-27-04459-f006:**
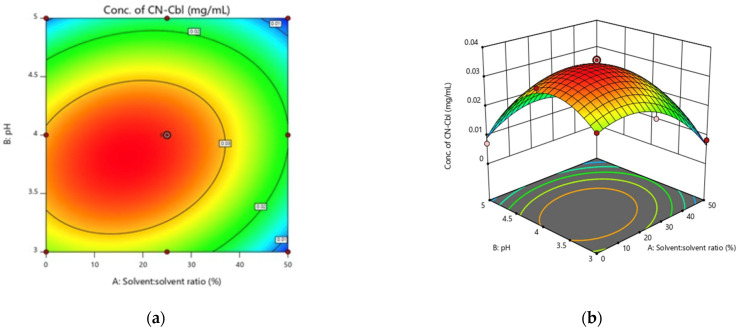
Surface response of the concentration of CN-Cbl recovered as a function of (**a**) AB interaction in contour plot and (**b**) AB interaction in the 3D surface plot.

**Figure 7 molecules-27-04459-f007:**
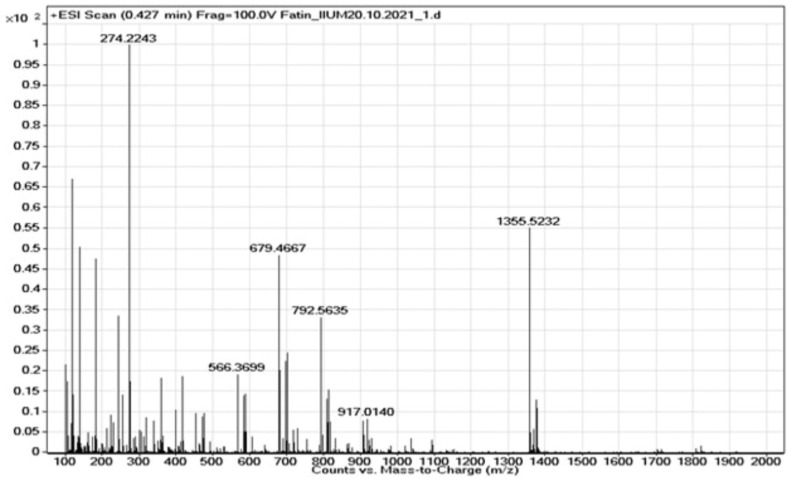
Mass spectrum of CN-Cbl in the purified oven-dried *U. lactuca* extract.

**Table 1 molecules-27-04459-t001:** Optimised design layout of CCD for the extraction of vitamin B12 from the *U. lactuca* oven-dried samples utilising the UAE technique.

Run	Solvent:Solvent Ratio (MeOH:H_2_O)	pH	Conc. of Vitamin B12 (mg/mL)
1	0:100	3	0.0250
2	50:50	3	0.0084
3	0:100	5	0.0072
4	50:50	5	0.0055
5	0:100	4	0.0323
6	50:50	4	0.0189
7	25:75	3	0.0225
8	25:75	5	0.0192
9	25:75	4	0.0357
10	25:75	4	0.0355
11	25:75	4	0.0351
12	25:75	4	0.0358
13	25:75	4	0.0305

**Table 2 molecules-27-04459-t002:** The ANOVA analysis of vitamin B12 extracted from the oven-dried *U. lactuca* samples that employed the UAE method.

Source	Sum of Squares	df	Mean Square	*F*-Value	*p*-Value	
Model	0.0003	5	0.0003	49.26	<0.0001	Significant
A	0.0002	1	0.0002	27.14	0.0012	
B	0.0001	1	0.0001	15.64	0.0055	
AB	0.0001	1	0.0001	9.06	0.0196	
A²	0.0002	1	0.0002	37.27	0.0005	
B²	0.0005	1	0.0005	85.86	<0.0001	
Residual	0.0000	7	6.179 × 10^−^^7^			
Lack of fit	0.0000	3	7.511 × 10^−^^6^	1.45	0.3538	Not significant
Pure error	0.0000	4	5.180 × 10^−^^6^			
Cor total	0.0016	12				

## Data Availability

Data is contained within the article or Appendix A.

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
