# Peer review of "Optimisation of Vitamin B12 Extraction from Green Edible Seaweed (Ulva lactuca) by Applying the Central Composite Design"

_molecules, 2022, doi:10.3390/molecules27144459_

Round 1

Reviewer 1 Report

This paper describes the extraction of Vitamin B12 from green edible seaweed (Ulva lactuca). The article is of scientific interest, but the authors must describe in greater detail the methods they have used, as well as delve into the description of the results obtained. Authors must take care of the formal aspects, fine-tune the scientific writing, extensively review the English writing, , etc. For this, I suggest that the article be completely revised by a professional proofreader in scientific writing and in English.

Mayor aspects

Extraction methods are not properly described.

In general, all materials and methods are not adequately described.

I am not at all clear about the homogeneity of the samples. Authors should use the same samples to compare extraction techniques.

Since vitamin B12 is photosensitive, I don't understand "air drying" or "sun drying".

The fact that the freeze-dried method does not give the best results must be due to the lack of homogeneity of the samples.

Materials and methods: Include the city and country of all the companies cited, and cite the companies of all the reagents and equipment’s employed. In case of USA companies, include the city and the state abbreviation. Unify and apply to the entire document.

I consider that there are very few optimized variables in the extraction methods used.

For all these reasons, I consider that the results obtained are not at all reliable.

Author Response

Dear Reviewer

We have addressed all the comments the file attached

Thank you for the comments and suggestion

Reviewer 2 Report

Manuscript ID: molecules-1773620 is focused on the optimisation of vitamin B12 extraction from green edible 2 seaweed Ulva Lactuca by applying the central composite 3 design.

The manuscript is well presented and discussed, but some points deserve revision (see below).

Please include the structural chemistry of vitamin B12 in the introduction to emphasize its structural aspects and their relationships with properties and function. 

More details are needed about sample collection and preparation. Macroalgae characteristics vary depending on the environmental conditions; it is necessary to specify:

Johor, Malaysia?

Latitude and longitude coordinates?

Water average temperatures in January?

Please include equipment details for the drying methods employed and for the extraction of vitamin B12.

Analytical standard stock solutions of vitamin B12 (Sigma Aldrich) were purchased from Merck. It is necessary to include more information about the companies.

Chromatographic conditions. Including the HPLC, detector, software, and column characteristics are mandatory.

Tandem Mass Spectrometry/Mass Spectroscopy (MS/MS). More detailed information about the equipment is needed.

Please include an HPLC chromatogram of vitamin B12 detection in the purified oven-dried U. Lactuca extract.

In Figure 5, please replace U. lactuca with U. Lactuca

The conclusion section must be improved. It is more like an abstract; please modify it.

Author Response

Dear Reviewer

We have addressed all the comments in the file attached

Thank you for the suggestion and comments

Round 2

Reviewer 1 Report

The authors have conveniently answered and/or modified some of the questions, but others still require improvement:

Resolution of figures should be improved.

Lines 110, 118, 138, 139,….: Put “p” in italics. Unify and apply to the entire document.

Section 2.5: The optimal real parameters must be obtained directly from the statistical software. It is neither necessary nor desirable to do this visually from a response surface plot.

Lines 256, 261,..: Put “m/z” in italics. Unify and apply to the entire document.

Materials and methods: Include the city and country of all the companies cited, and cite the companies of all the reagents and equipment’s employed. In case of USA companies, include the city and the state abbreviation. Unify and apply to the entire document. The authors have tried to modify this aspect, but in a very poor way.

Section 4.1.: The authors must describe how they guarantee the homogeneity of the samples that are later used in the different drying processes. It is not described at all, and it gives me many doubts.

Lines 374-376: Amount of sample? Volume? Where it has boiled?. Describe in a much deeper way.

Lines 377-380: Centrifugued at what g? Describe the centrifuge. Describe in a much deeper way.

Lines 381-384: Temperature? Potency? Cycle?. Describe in a much deeper way.

Section 4.3.: How much volume of each solvent passes through the column? Describe in a much deeper way.

Section 4.4.3.: 100% MeOH with 0.1% acetic acid for an isocratic gradient? Check that it is so. It seems to me that such an amount of MeOH is not feasible.

Section 4.7.: 100% MeOH with 0.1% acetic acid for an isocratic gradient? Check that it is so. It seems to me that such an amount of MeOH is not feasible.

References 8, 10, 14, 22, 23, 26: Information is missing or the format is not the correct. Revise all the references.

Author Response

Thank you for the comments, we really appreciate it.  Kindly find the response int the file attached

Reviewer 2 Report

The manuscript has been improved, and the reviewer's suggestions have been incorporated. Nevertheless, all the paper figures must be improved; they present poor definitions in the present form. 

Author Response

Thank you for the comments, we really appreciate it. Kindly find our response in the attachment
